META-RESEARCH ARTICLE

# Language, economic and gender disparities widen the scientific productivity gap

Tatsuya Amano[1,2,*], Valeria Ramírez-Castañeda[3,4], Violeta Berdejo-Espinola[1,2], Israel Borokini[5], Shawan Chowdhury[6], Marina Golivets[7], Juan David González-Trujillo[8], Flavia Montaño-Centellas[9,10], Kumar Paudel[11], Rachel L. White[12], Diogo Veríssimo[13]

1 School of the Environment, The University of Queensland, Brisbane, Queensland, Australia, 2 Centre for Biodiversity and Conservation Science, The University of Queensland, Brisbane, Queensland, Australia, 3 Museum of Vertebrate Zoology, University of California, Berkeley, California, United States of America, 4 Department of Integrative Biology, University of California, Berkeley, California, United States of America, 5 Department of Ecology, Montana State University—Bozeman, Bozeman, Montana, United States of America, 6 School of Biological Sciences, Monash University, Melbourne, Victoria, Australia, 7 Department of Community Ecology, Helmholtz Centre for Environmental Research—UFZ, Halle, Germany, 8 Departamento de Biología, Facultad de Ciencias, Universidad Nacional de Colombia, Sede Bogotá, Colombia, 9 Instituto de Ecología, Universidad Mayor de San Andrés, La Paz, Bolivia, 10 Department of Biological Sciences, Louisiana State University, Baton Rouge, Louisiana, United States of America, 11 Greenhood Nepal, Kathmandu, Nepal, 12 School of Applied Sciences, University of Brighton, Brighton, United Kingdom, 13 Department of Biology, University of Oxford, Oxford, United Kingdom

* t.amano@uq.edu.au

## Abstract

Scientific communities need to understand and eliminate barriers that prevent people with diverse backgrounds from contributing to and participating in science. However, the combined impact of individuals' linguistic, economic, and gender backgrounds on their scientific productivity is poorly understood. Using a survey of 908 environmental scientists, we show that being a woman is associated with up to a 45% reduction in the number of English-language publications, compared to men. Being a woman, a non-native English speaker, and from a low-income country is associated with up to a 70% reduction, compared to male native English speakers from a high-income country. The linguistic and economic productivity gap narrows when based on the total number of English- and non-English-language publications. We call for an explicit effort to consider linguistic, economic, and gender backgrounds and incorporate non-English-language publications when assessing the performance and contribution of scientists.

## Introduction

Currently, not everyone can contribute to science in an equal manner due to a number of barriers. This is a serious equity issue in science, as all scientists, regardless of their background, should have an equal opportunity to contribute to science, as

**Data availability statement:** The model-based estimates underlying Figs 1A–1F and 2A–2D can be found in S1 Data. The raw data plotted in Fig 1A–1F are from the survey questions, which our ethics approval prevents us from sharing to secure confidentiality of the respondents. Access to these raw data should be requested from the University of Queensland Ethics office, which can be contacted at humanethics@ research.uq.edu.au. All codes used in the analysis are available at: https://osf.io/b3cza.

**Funding:** This work was supported by the following grants: Australian Research Council Future Fellowship FT180100354 (TA, V.B.-E.), Australian Research Council Discovery Project DP230101734 (TA, V.B.-E.), University of Queensland strategic funding (TA), and German Research Foundation (DFG-FZT 118, 202548816) (SC). The funders had no role in study design, data collection and analysis, decision to publish, or preparation of the manuscript.

**Competing interests:** The authors have declared that no competing interests exist.

**Abbreviations:** ARC, Australian Research Council; DORA, Declaration on Research Assessment; ROPE, Research Opportunity and Performance Evidence.

stated in the UNESCO Recommendation on Open Science [1]. These barriers also deprive the scientific community of the diversity of people, ideas, and approaches that are key to innovation in science and to addressing ongoing global challenges [2–6]. Therefore, the scientific community urgently needs to understand and dismantle the barriers to scientists, particularly those from historically and currently underrepresented groups.

Many factors other than one's own abilities can affect the performance, recognition, and representation of scientists. For example, women publish fewer articles [7,8], attract fewer citations [9], are less successful in grant applications [8], win a lower proportion of awards [10], are under-represented as journal editors [11], patent at a lower rate [12], perform more teaching [13] and internal services [14], are less likely to hold a tenured position [15], and more likely to leave academia [16] than men. Women, non-binary individuals, and people of color are more vulnerable to the negative impact of unprofessional peer reviews on their careers [17]. Scientists from lower-income countries also publish fewer articles [18], receive less favorable review outcomes [19], are less funded [20], and face more barriers when traveling for academic purposes [21] than those from higher-income countries. Non-native English speakers spend more time when conducting scientific activities and disseminating research [22] and find their science rated lower [19,23] than native English speakers, and tend to suffer from dissatisfaction, anxiety [24], and imposter syndrome [25].

Few studies to date, however, have assessed the relative and combined impacts of gender, linguistic, and socio-economic backgrounds on scientific productivity by individual scientists. For instance, the difference in scientific productivity has been tested extensively among gender identities [7], but rarely between native and non-native English speakers. This is likely because it is almost impossible to collect accurate information on the linguistic background of authors in large bibliometric studies, while survey-based studies tend to be targeted at a single country or focused only on non-native English-speaking scientists. Scientific productivity, typically measured by the number of English-language publications, is still widely used to evaluate the performance of scientists, although its validity is often questioned [26]. We urgently need to assess which attributes of scientists other than gender identities influence their productivity, to understand how not accounting for those attributes can bias the common metric of scientific performance, further disadvantaging the careers of scientists from underrepresented groups.

This study capitalizes on a survey of 908 environmental scientists from eight nationalities to test how the productivity of scientists differs depending on their gender, linguistic, and economic backgrounds. This dataset has three major advantages: the survey (i) covers participant nationalities with varying levels of English proficiency and income, (ii) records the self-reported first languages of participants, and (iii) measures the scientific productivity in terms of the number of English and non-English-language publications for scientists with 1–55 years in their careers. This allows us to compare the relative effect of participants' gender identity, first language, and economic backgrounds, and their combined impacts, on the number of their publications in English and in non-English languages across different career stages.

## Results and discussion

The survey collected responses from 908 researchers in environmental sciences from eight nationalities each with varying levels of English proficiency and income [22]: Bangladeshi ($n = 106$), Bolivian (100), British (112), Japanese (294), Nepali (82), Nigerian (40), Spanish (108), and Ukrainian (66) (See Materials and methods for more detail and S1 Table for the number of participants by English proficiency, income level, and gender identity).

We found that women, non-native English speakers, and those from lower-income countries published statistically fewer English-language peer-reviewed papers than men, native English speakers, and those from higher-income countries, respectively, when controlling for the number of years in research and their disciplines (Fig 1A–1C, Table 1). The male–female productivity gap was especially wide in early career researchers (Fig 1A), although the interaction term was not statistically significant (Table 1). The gender-other interaction term was significant (Table 1), however, the small sample size of the gender-other category (e.g., only two in English native, S1 Table) makes it difficult to conclude whether this is a real pattern or a statistical artifact. In contrast, the significant interaction term for those with low English proficiency indicates that the language productivity gap was wider in scientists at a later career stage (Table 1, Fig 1B). The interaction between the number of years in research and income level was not significant, indicating that the income productivity gap did not differ between participants with different levels of research experience (Fig 1C and Table 1).

The results were in stark contrast when we ran the same analysis but using the total number of English- and non-English-language papers as a measure of productivity. Non-native English speakers at early to mid-career stages published statistically more peer-reviewed papers in English and non-English languages combined, than native English speakers (Fig 1E and Table 2). The income productivity gap was also reversed; those from lower-income countries published a statistically higher total number of peer-reviewed papers than those from higher-income countries (Fig 1F and Table 2). Women still published less than men even when the analysis was based on the number of papers in English and non-English languages combined (Fig 1D and Table 2).

The analysis above used the level of countries' English proficiency to approximate the level of each participant's English proficiency. To further test the potential role of individuals' levels of English proficiency, we also conducted a separate analysis focusing only on non-native English-speaking participants. In this analysis, we included an additional explanatory variable—the number of years spent living in countries where English is the first language—as more exposure to English is known to be correlated with higher English proficiency [27,28]. We found that non-native English speakers who have lived longer in English-speaking countries published more peer-reviewed papers in English (S2 Table). Although the number of years spent living in countries where English is the first language can also be associated with other factors, such as access to collaboration, this result indicates that scientific productivity in English varies even among non-native English speakers, and can be explained partly by the individuals' level of English proficiency.

These results provide clear evidence that language, economic, and gender disparities widen the scientific productivity gap, particularly when focusing only on English-language publications. This is likely due to the numerous barriers that women and non-binary people, non-native English speakers, and those from lower-income countries experience when conducting science [8,17,19,21,22,29,30]. For example, non-native English speakers spend up to 51% more time to write a paper in English, experience paper rejection due to English writing up to 2.6 times more often, and are requested to improve their English writing during paper revision up to 12.5 times more often, compared to native English speakers [22]. Our findings are based on regression analyses, and thus, may not necessarily indicate causation. Nevertheless, when the total number of English- and non-English-language papers was used as a measure of scientific productivity, we found no or even reversed productivity gap between non-native English speakers and native English speakers, and between lower-middle-income and high-income countries. This gives a strong signal that the need to publish papers in a language that is not their first language, which also often demands considerable cost [31], has led to fewer English-language

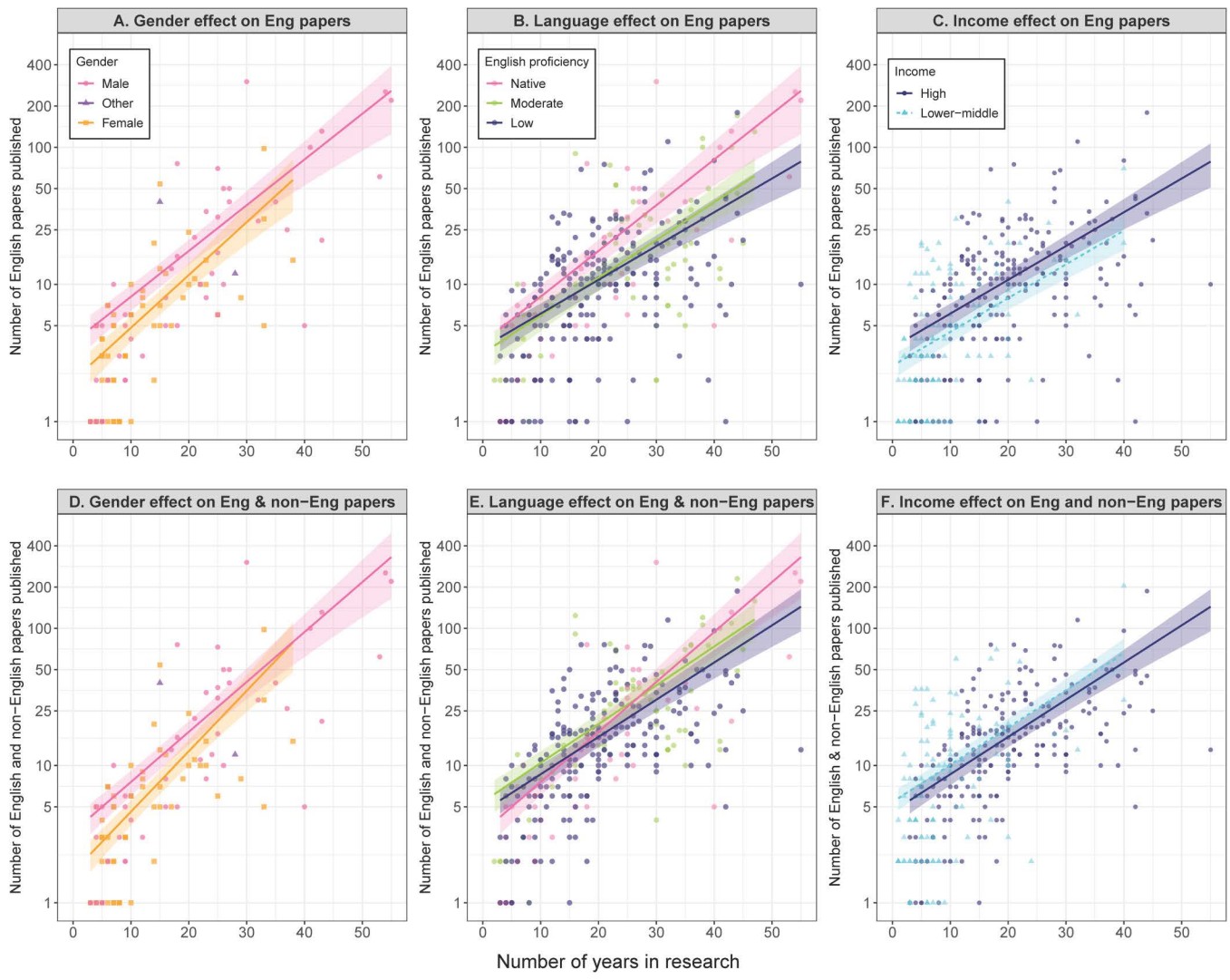

**Fig 1. Impact of gender, language, and economic backgrounds on scientific productivity. (A)** Gender, **(B)** language, and **(C)** income effects on the number of English-language papers published by participants with varying number of years in research. **(D)** Gender, **(E)** language, and **(F)** income effects on the number of English- and non-English-language papers published by participants. Although all samples ($n = 908$) were used to estimate the coefficient of each explanatory variable, each panel only displays those samples that are relevant to the comparison of focus, i.e., (A, D) native English speakers with all gender categories from a high-income country, (B, E) male participants with all language backgrounds from high-income countries, and (C, F) male participants with low English proficiency from high or lower-middle income countries. The regression lines (with 95% confidence intervals as shaded areas) represent the estimated relationship based on the results shown in Tables 1 and 2. The data underlying this figure can be found in S1 Data.

publications from non-native English speakers and those from lower-income countries. As a result, these scientists are portrayed as less productive based on English-language publication metrics.

To further visualize the accumulated impact of linguistic, economic, and gender backgrounds of individual researchers on their scientific productivity, we used the models developed in Tables 1 and 2 to estimate the expected absolute and percentage difference in scientific productivity between researchers with different combinations of the three attributes (Fig 2). When using the number of English-language peer-reviewed papers published as a measure of productivity, being a woman alone was associated with, on average, a reduction in the number of peer-reviewed publications at a late career

**Table 1. Results of a generalized linear model (with a negative binomial distribution) of factors explaining variations in the number of English-language peer-reviewed papers published by survey participants ($n = 908$). Number of years in research was centered before the analysis. The reference category for English proficiency, Income level, Gender, and Discipline was English native, High income, Male, and Conservation biology, respectively. Significant results are shown in bold. The bias-adjusted estimate of mean-square error (used as a predictive measure) based on the 10-fold cross-validation of the final model is 474.51, representing a 13.46% improvement from the null model with 548.34.**

| Coefficients | Estimate | Standard error | z | p |
|---|---|---|---|---|
| Intercept | 2.53 | 0.10 | | |
| **Number of years in research** | **0.077** | **0.0063** | **12.27** | **$<0.20 \times 10^{-15}$** |
| **English proficiency—low** | **−0.40** | **0.083** | **−4.81** | **$1.48 \times 10^{-6}$** |
| **English proficiency—moderate** | **−0.39** | **0.096** | **−4.10** | **$4.08 \times 10^{-5}$** |
| **Income level—lower-middle** | **−0.31** | **0.069** | **−4.43** | **$9.63 \times 10^{-6}$** |
| Gender—other | −0.0040 | 0.27 | −0.015 | 0.99 |
| **Gender—female** | **−0.45** | **0.065** | **−7.01** | **$2.33 \times 10^{-12}$** |
| **Discipline—ecology** | **0.24** | **0.085** | **2.81** | **0.0050** |
| **Discipline—evolutionary biology** | **0.22** | **0.11** | **1.97** | **0.049** |
| **Discipline—other** | **0.33** | **0.11** | **2.87** | **0.0041** |
| **Discipline—other biological sciences** | **0.22** | **0.10** | **2.24** | **0.025** |
| **Number of years in research × English proficiency—low** | **−0.020** | **0.0072** | **−2.78** | **0.0055** |
| Number of years in research × English proficiency—moderate | −0.013 | 0.0076 | −1.75 | 0.080 |
| **Number of years in research × Gender—other** | **−0.059** | **0.022** | **−2.64** | **0.0083** |
| Number of years in research × Gender—female | 0.012 | 0.0065 | 1.81 | 0.070 |
| **Variables removed based on the likelihood ratio test** | **$\chi^2$** | **p** | | |
| Number of years in research × Income level | 0.10 | 0.75 | | |

**Table 2. Results of a generalized linear model (with a negative binomial distribution) of factors explaining variations in the number of English- and non-English-language peer-reviewed papers combined, published by survey participants ($n = 908$). Number of years in research was centered before the analysis. The reference category for English proficiency, Income level, Gender, and Discipline was English native, High income, Male, and Conservation biology, respectively. Significant results are shown in bold. The bias-adjusted estimate of mean-square error (used as a predictive measure) based on the 10-fold cross-validation of the final model is 647.49, representing a 15.29% improvement from the null model with 764.39.**

| Coefficients | Estimate | Standard error | z | p |
|---|---|---|---|---|
| Intercept | 2.50 | 0.097 | | |
| **Number of years in research** | **0.084** | **0.0061** | **13.72** | **$<0.20 \times 10^{-15}$** |
| English proficiency—low | 0.0074 | 0.080 | 0.092 | 0.93 |
| **English proficiency—moderate** | **0.21** | **0.091** | **2.31** | **0.021** |
| **Income level—lower-middle** | **0.16** | **0.065** | **2.43** | **0.015** |
| Gender—other | 0.42 | 0.25 | 1.68 | 0.092 |
| **Gender—female** | **−0.40** | **0.061** | **−6.55** | **$5.83 \times 10^{-11}$** |
| Discipline—ecology | 0.083 | 0.079 | 1.05 | 0.29 |
| Discipline - evolutionary biology | −0.066 | 0.10 | −0.64 | 0.53 |
| Discipline—other | 0.15 | 0.11 | 1.40 | 0.16 |
| Discipline—other biological sciences | 0.085 | 0.093 | 0.91 | 0.37 |
| **Number of years in research × English proficiency—low** | **−0.021** | **0.0070** | **−3.05** | **0.0023** |
| **Number of years in research × English proficiency—moderate** | **−0.019** | **0.0074** | **−2.54** | **0.011** |
| Number of years in research × Gender—other | −0.028 | 0.021 | −1.36 | 0.17 |
| **Number of years in research × Gender—female** | **0.018** | **0.0062** | **2.85** | **0.0044** |

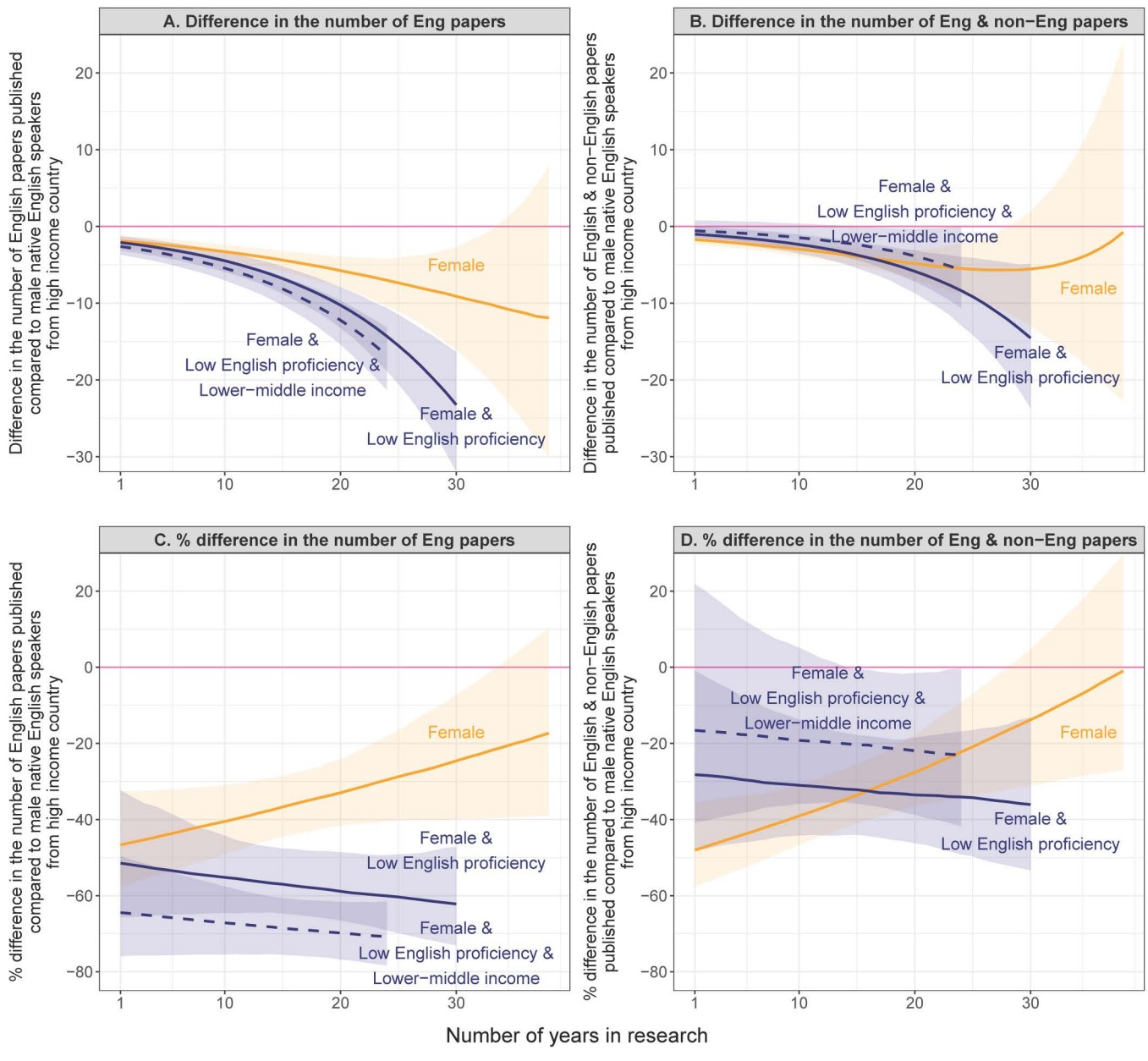

**Fig 2. Additive disadvantages of being a woman with low English proficiency and from a low-income country in scientific productivity. (A)** Absolute difference in the number of English-language peer-reviewed papers published between male native English speakers from a high-income country (baseline shown in pink) and female native English speakers from a high-income country (solid line in orange), female non-native English speakers from a high-income country (solid line in navy), and female non-native English speakers from a lower-middle income country (dashed line in navy). Here, non-native English speakers are defined as those with low English proficiency. **(B)** Absolute difference in the number of English- and non-English-language peer-reviewed papers published between researchers with the same combinations of the three attributes as (A). **(C)** Percentage difference in the number of English-language peer-reviewed papers published between researchers with the same combinations of the three attributes as (A). **(D)** Percentage difference in the number of English- and non-English-language peer-reviewed papers published between researchers with the same combinations of the three attributes as (A). The lines (with 95% confidence intervals as shaded areas) represent median estimates. The data underlying this figure can be found in S1 Data.

stage by over 10 compared to men, while being a woman and a non-native English speaker equated to a 20 or more reduction in peer-reviewed publications at a late career stage, compared to male native English speakers (Fig 2A). The relative productivity impact of being a woman was largest at an early career stage, with over 45% reduction in the number of publications compared to men, while the relative impact was reduced at a later career stage, with about 20% reduction in the number of publications (Fig 2C). The relative productivity impact of being a non-native English speaker and from a lower-income country was largest at a late career stage. Being a woman and a non-native English speaker equated to up to a 60% reduction, and being a woman, a non-native English speaker, and from a lower-income country was associated with even a 70% reduction in the number of publications (Fig 2C)

The linguistic and economic productivity gap persisted but clearly narrowed, when using the total number of English- and non-English-language papers published as a measure of scientific productivity (Fig 2B and 2D). In absolute terms, being a woman and a non-native English speaker equated to a reduction of up to 15 publications on average (Fig 2B), rather than over 20 (Fig 2A), compared to male native English speakers. Being a woman, a non-native English speaker, and from a lower-income country equated to a reduction of up to five (Fig 2B), rather than 15 (Fig 2A) publications, compared to male native English speakers from a high-income country. The additive impact of being a woman, a non-native English speaker, and from a lower income country was also drastically reduced in relative terms when taking non-English-language publications into account, with the productivity gap between female non-native English speakers and male native English speakers narrowing to up to 30% (Fig 2D), rather than over 60% (Fig 2C), and the productivity gap between female non-native English speakers from a lower income country and male native English speakers from a high-income country falling to over 20% (Fig 2D), rather than 70% (Fig 2C).

These results provide robust evidence that the impact of each of the three attributes (gender, linguistic, and economic background) adds up to create an almost insurmountable disadvantage, especially for female non-native English speakers from lower-income countries, in contributing to and participating in science. Being a woman alone was associated with a considerable disadvantage in terms of productivity, especially at an early career stage, with the number of publications almost halving compared to male counterparts. The larger gender productivity gap at an earlier career stage is likely due to multiple disadvantages for early-career women, such as a higher rate of taking a career break due to parental, family, and caring responsibilities [32], larger impact of parenthood [33], and less involvement in collaborations [34] compared to men.

Being a non-native English speaker is associated with a further 15% reduction, and being from a lower-income country equates to an additional 10% reduction in publications. The productivity impact of being a non-native English speaker and from a lower-income country was larger for those at a later career stage. A potential explanation for this is the Matthew Effect; scientists who have previously been successful are more likely to succeed again in the future, causing differences in future success between winners and non-winners to further grow as their career progresses [35,36]. This indicates that the language and economic disparity may have a cumulative and long-lasting impact on scientists' productivity over their careers. It may also be explained, for example, by the recent increase in pressure on early-career researchers to publish in English, even in countries where English is not widely spoken [37], or by the tendency of early-career researchers to leverage emerging artificial intelligence technologies more to boost their productivity [38]. It is worth emphasizing, however, that non-native English speakers at an early career stage still publish less in English than their native English-speaking counterparts (Figs 1B and 2C).

Our study may potentially be underestimating the productivity impact of the gender, linguistic, and economic backgrounds of scientists, both at earlier and later career stages. First, our analysis excluded researchers without any first-authored peer-reviewed paper in English, which might have caused the underestimation of the productivity gap at an early career stage. Second, the survey that produced the data used in this study is unlikely to have included participants who have discontinued their scientific careers (see Materials and methods), potentially creating survivorship bias in the results. To fully understand the impact of the three attributes on scientific productivity, future longitudinal research

needs to scrutinize differences in career trajectories between those with different linguistic and economic backgrounds, as has been done for gender identities [16]. Qualitative causal inference approaches, such as process tracing and general elimination methodology, would also be useful to better understand the mechanisms underlying the patterns found in this study. Future studies should also assess the association between such attributes and other metrics of scientific productivity, such as authorship positions, journals for publications, and citations. We also recognize that the categories of gender, language and economic background used in this study are coarse, as also reflected in the modest improvements in the predictive measure of the final model from the null model (Tables 1 and 2, and S2 Table), and more detailed background information, such as more detailed gender identities, or individuals' levels of English proficiency and income, may further explain the variation in productivity. Other factors that were not collected in the survey, such as job titles and the type of institutions, may also explain the variation in productivity. The survey data used in this study are limited only to researchers in environmental sciences and within the eight nationalities. We call for further research to assess whether similar patterns would hold in other disciplines and countries with different publishing cultures and collaboration norms.

The results of this study have implications for how we should assess an individual scientist's productivity in research assessment. Despite the increasing tendency to diversify the criteria used to assess an individual scientist's contributions in, for example, hiring, promotion, or funding decisions (Declaration on Research Assessment (DORA): https://sfdora.org/), the number of publications in English, together with other publication metrics, is still widely used in research assessment. The combined impact of gender, linguistic, and economic backgrounds of individual scientists is rarely taken into account. For example, the Australian Research Council (ARC) has introduced the Research Opportunity and Performance Evidence (ROPE) policy to allow researchers to declare significant interruptions that have affected their research capacity, productivity, or contribution in the National Competitive Grant Program [39]. Nevertheless, examples of "significant interruptions" proposed by the ARC only include interruptions to academic employment, disasters, misadventure, medical conditions, disability, caring and parental responsibilities, and community obligations [39], leaving out the considerable disadvantages associated with individuals' linguistic and economic backgrounds. Our findings suggest that being a non-native English speaker and from a lower-income country also should be a factor that is considered explicitly in any research assessment as a major impediment to the research capacity, productivity, and contribution of scientists.

The scientific community also largely ignores non-English-language publications in research assessment, even in countries where English is not widely spoken [40]. Our results indicate that this common practice could further exacerbate the disadvantages of non-native English speakers and those from lower-income countries. Non-English-language publications can also be an important source of evidence, based on a robust study design, to inform decisions in addressing global challenges, such as the biodiversity and climate crises [41,42]. Including non-English-language publications in research assessment by individuals, institutions, and funders, which also conforms with the DORA's emphasis on what is published rather than where it is published, can also reduce, though not eliminate, the impact of linguistic and economic disadvantages in science.

Our findings indicate a clear need to understand the cumulative impact of having multiple attributes that can disadvantage a scientist, not only on the number of publications, but more broadly on the contribution, performance, and representation of individual scientists. Recent studies on gender inequality in science point the way forward; we already know how gender impacts scientific productivity [7], citations [9], funding success [8], employment [43], promotion [44], representation [11], and so on. As science is becoming increasingly globalized, individual scientists' attributes other than gender identity, most notably, but not limited to, linguistic and economic backgrounds, also form the fundamental basis of diversity in science. We urge the scientific community to assess the cumulative disadvantage faced by currently and historically underrepresented groups in science, and take actions to achieve their full contribution to and fair participation in science. Quantifying the impact of these barriers alone would not solve the issue. However, those who are not directly affected by the barriers cannot easily visualize their impacts. Therefore, as an initial step towards addressing these barriers, we need to try and build a consensus within the scientific community about the impact of various barriers by generating and presenting the evidence.

## Materials and methods

### Ethics statement

The survey obtained the University of Queensland's Institutional Human Research Ethics Approval (committee: Science Low and Negligible Risk Committee, approval number: 2021/HE000566). All participants were over 18 years old and agreed to participate in the survey through written consent. The survey provided the Participant Information Sheet that clarifies the voluntary nature of participation, the aims of the research, how the data would be used, and that all data would be confidential.

### Data

The data used in this study was collected by a survey published in another study [22]. The survey was conducted between June and October 2021, with the aim of quantifying the amount of effort needed by individual researchers with different linguistic and economic backgrounds to conduct scientific activities in English and their first language (see [22] for more details of the survey). The survey was targeted at eight nationalities: Bangladeshi, Bolivian, British, Japanese, Nepali, Nigerian, Spanish, and Ukrainian. These nationalities were selected based on the levels of each country's English proficiency (based on the English Proficiency Index [45]) and income (based on the World Bank list of economies [46]): Bangladeshi, Nepali (low English proficiency and lower-middle income), Japanese (low English proficiency and high income), Bolivian, Ukrainian (moderate English proficiency and lower-middle income), Spanish (moderate English proficiency and high income), Nigerian (English as an official language and lower-middle income), and British (English as an official language and high income). Anyone who has one of the selected nationalities and has published at least one first-authored peer-reviewed English-language paper in ecology, evolutionary biology, conservation biology, or related disciplines was eligible to participate in the survey, regardless of their career level or profession.

The survey was initially developed in English, translated into the relevant languages for each nationality (Bangla for Bangladeshi, Japanese for Japanese, Nepali for Nepali, Spanish for Bolivian and Spanish, and Ukrainian for Ukrainian) and distributed in each of the eight countries in as unbiased a way as possible, through major mailing lists, and/or academic societies, universities, and institutions of relevant disciplines, or directly to relevant researches who were systematically identified on literature search systems. Using personal networks was avoided to reduce potential biases in participant recruitment (see [22] for more details of the survey distribution). Due to this nature of survey distribution, the survey was largely limited to those researchers who were active in their careers at the time of the survey, and unlikely to include those who had already discontinued their scientific careers.

The survey was answered by a total of 908 researchers in environmental sciences (mostly ecology, evolutionary biology, conservation biology, and related disciplines) with at least one first-authored peer-reviewed paper in English. The number of participants with each nationality was as follows: Bangladeshi ($n = 106$), Bolivian (100), British (112), Japanese (294), Nepali (82), Nigerian (40), Spanish (108), and Ukrainian (66). The gender composition of the participants was 339 female, 556 male, and 13 participants in other categories, with the median age of 39 (range: 18–77) years old and median 13 (range: 1–55) years of experience in research. See S1 Table for the number of participants by English proficiency, income level, and gender identity.

### Statistical analysis

We first performed a generalized linear model assuming a negative binomial distribution, with the number of English-language peer-reviewed papers published by survey participants as the response variable, and five explanatory variables: the number of years in research (centered), a country's English language proficiency (English native as the reference category, moderate, and low), a country's income level (high as the reference category, and lower-middle), the gender identity of the participant (male as the reference category, female, and other), and the research discipline of the participant

(conservation biology as the reference category, ecology, evolutionary biology, other biological sciences, and other). We could not control for nationalities, as the variable was almost perfectly collinear with English proficiency and income levels. We also included three interactions: the number of years in research and a country's English language proficiency, the number of years in research and a country's income level, and the number of years in research and the gender identity of the participant. We first tested whether the three interactions were significant using the likelihood ratio test and found that the interaction between the number of years in research and a country's income level was not significant (Table 1). We therefore removed this non-significant interaction from all analyses. After removing this interaction, we confirmed that a country's income level itself was significant based on the likelihood ratio test and decided to keep this explanatory variable in the final model. We interpreted the results derived from the final model.

We next fitted the same model as the final model in the first analysis, but using the total number of English- and non-English-language peer-reviewed papers published by participants as the response variable. Lastly, we fitted the same model as the final model in the first analysis, but excluding native English speaking participants and including the number of years spent living in countries where English is the first language as an additional explanatory variable.

To assess the predictive performance of the three final models, we also performed 10-fold cross-validation for each final model and the null model (i.e., the model only with the intercept) and calculated a bias-adjusted estimate of the mean-square error of each model as a predictive measure [47].

We then used the models developed in the first and second analyses (shown in Tables 1 and 2, respectively) to estimate the expected absolute and percentage difference in scientific productivity between male native English speakers from a high-income country (baseline) and (i) female native English speakers from a high-income country (representing the effect of being a female), (ii) female non-native English speakers from a high-income country (representing the effect of being a female non-native English speaker), and (iii) female non-native English speakers from a lower-middle income country (representing the effect of being a female non-native English speaker from a lower-middle income country). Here, non-native English speakers were defined as those with low English proficiency.

For each of the seven coefficients that are necessary for the calculation (intercept, the number of years in research, English proficiency—low, income level—lower-middle, gender—female, the number of years in research × English proficiency—low, and the number of years in research × gender—female), we derived 1,000 coefficient estimates from a normal distribution with the mean of the estimated coefficient and s.d. of the standard error of the coefficient in each model. We used the 1,000 sets of coefficient estimates to calculate 1,000 estimates of the expected number of (i) English-language peer-reviewed papers and (ii) English-language and non-English-language peer-reviewed papers combined, for (a) a male native English speaker from a high-income country (with a varying number of years in research between 1 and 38 years), (b) a female native English speaker from a high-income country (between 1 and 38 years), (c) a female non-native English speaker from a high-income country (between 1 and 30 years), and (d) a female non-native English speaker from a lower-middle income country (between 1 and 24 years). The year range used was the actual year range for the participants of the respective groups. We then calculated the absolute and percentage differences between (a) and (b), (c), and (d), respectively, and used the median and 2.5th and 97.5th percentiles of the 1,000 estimates to plot the results. The estimates assumed the reference category (conservation biology) for discipline. We decided not to estimate the effect of gender—other due to the small sample size (13 participants, S1 Table).

The analysis was conducted using R version 4.4.0 [48] and the following R packages: tidyverse [49], MASS [50], lmtest [51], gridExtra [52], and cv [53].

## Supporting information

**S1 Table. Number of survey participants by English proficiency, income level, and gender identity.**
(DOCX)

**S2 Table. Results of a generalized linear model (with a negative binomial distribution) of factors explaining variations in the number of English-language peer-reviewed papers published by survey participants whose first language is not English (*n* = 754).** Survey participants whose first language is English were excluded from this analysis. Number of years in research was centered before the analysis. The reference category for English proficiency, Income level, Gender, and Discipline was Low English proficiency, High income, Male, and Conservation biology, respectively. Significant results are shown in bold. The bias-adjusted estimate of mean-square error (used as a predictive measure) based on the 10-fold cross-validation of the final model is 240.55, representing a 24.26% improvement from the null model with 317.61.
(DOCX)

**S1 Data. The data underlying Figs 1A–1F and 2A–2D.**
(XLSX)

## Acknowledgments

We thank all participants in the survey and M. Amano for English proofreading.

## Author contributions

**Conceptualization:** Tatsuya Amano, Valeria Ramírez-Castañeda, Diogo Veríssimo.

**Formal analysis:** Tatsuya Amano.

**Funding acquisition:** Tatsuya Amano, Shawan Chowdhury.

**Investigation:** Tatsuya Amano, Violeta Berdejo-Espinola, Israel Borokini, Shawan Chowdhury, Marina Golivets, Juan David González-Trujillo, Flavia Montaño-Centellas, Kumar Paudel, Rachel L. White.

**Methodology:** Tatsuya Amano, Valeria Ramírez-Castañeda, Diogo Veríssimo.

**Project administration:** Tatsuya Amano, Violeta Berdejo-Espinola.

**Validation:** Tatsuya Amano, Violeta Berdejo-Espinola.

**Visualization:** Tatsuya Amano.

**Writing – original draft:** Tatsuya Amano.

**Writing – review & editing:** Tatsuya Amano, Valeria Ramírez-Castañeda, Violeta Berdejo-Espinola, Israel Borokini, Shawan Chowdhury, Marina Golivets, Juan David González-Trujillo, Flavia Montaño-Centellas, Kumar Paudel, Rachel L. White, Diogo Veríssimo.

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
