## [Editor Report · Decision Letter 0]

17 Jan 2025

Dear Tatsuya,

Thank you for submitting your manuscript entitled "Language, economic, and gender disparities widen the scientific productivity gap" for consideration as a Meta-Research Article by PLOS Biology. Please accept my apologies for the slow process as we recovered from the journal office closure over the holidays.

Your manuscript has now been evaluated by the PLOS Biology editorial staff, as well as by an academic editor with relevant expertise, and I'm writing to let you know that we would like to send your submission out for external peer review.

Once your full submission is complete, your paper will undergo a series of checks in preparation for peer review. After your manuscript has passed the checks it will be sent out for review. To provide the metadata for your submission, please Login to Editorial Manager (https://www.editorialmanager.com/pbiology) within two working days, i.e. by Jan 21 2025 11:59PM.

Kind regards,

Roli

Roland Roberts, PhD

Senior Editor

PLOS Biology

rroberts@plos.org

---

## [Decision Letter · Decision Letter 1]

31 Mar 2025

Dear Tatsuya,

Thank you for your patience while your manuscript "Language, economic, and gender disparities widen the scientific productivity gap" went through peer-review at PLOS Biology. Your manuscript has now been evaluated by the PLOS Biology editors, an Academic Editor with relevant expertise, and by three independent reviewers. Please accept my apologies for the unusual delay in securing a decision for you.

You'll see that reviewer #1 is positive, and has a number of points that can be addressed textually (clarifications, limitations). Reviewer #2 is also very positive, and has a few suggestions, some of which might involve further analysis (e.g. country as fixed effect). Reviewer #3 is more cautious, and wants you to consider a number of alternative explanations and limitations, plus several analyses (with some overlap with rev #2), including the use of alternative metrics of “success.”

In light of the reviews, which you will find at the end of this email, we are pleased to offer you the opportunity to address the comments from the reviewers in a revision that we anticipate should not take you very long. We will then assess your revised manuscript and your response to the reviewers' comments with our Academic Editor aiming to avoid further rounds of peer-review, although we might need to consult with the reviewers, depending on the nature of the revisions.

**IMPORTANT - SUBMITTING YOUR REVISION**

*Resubmission Checklist*

*Published Peer Review*

*PLOS Data Policy*

*Blot and Gel Data Policy*

Sincerely,

Roli

Roland Roberts, PhD

Senior Editor

PLOS Biology

rroberts@plos.org

REVIEWERS' COMMENTS:

Reviewer #1:

[identifies herself as Antica Culina]

I enjoyed reading this manuscript. The topic is very relevant and timely, as it is becoming clear that the current evaluation system in science is not promoting diversity. This study uses a survey approach to quantify the gender, language, and economic correlates of scientific productivity measured using the most common measure of productivity - scientific publications.

I only have a few comments:

1. I am not sure if the wording 'to their full potential' is meaningful. What would a full potential be? Would it be potential that one would reach if one would just do science without any restrictions? I think rather than referencing one's work in respect to some full potential standard, it needs to be highlighted that the output of researchers is different, and this is depended on many factors, some of which cannot be changed. And thus the person has reached their full potential given those circumstances. My full potential is inevitably different than a full potential of someone else, and I cannot change it, but those who evaluate me can take the circumstances into account when evaluating me. Or take different metrics.

2. Line 65 - unclear meaning of 'apparent'.

3. I suggest adding some basic summary info on the survey at the beginning of results, such as number of responses per country etc. This would facilitate putting the results into context.

4. Lines 88-90 - it is unclear if this sentence compares woman vs man, non-native vs native speakers etc, or all the former group vs all of the latter group.

5. Unclear meaning of the sentence lines 98-99

6. Discussion currently lists one reasons why this survey might have underestimated the productivity gap. However, if I read the Methods correctly, I see two more reasons. First, the survey was in English (is this correct?), so probably responded by people who speak English. Those that do not speak English have likely not answer it. Second, emails were sent to only those that have first authored at least one publication in English, which means that those that do not have such authorship were excluded, which is also likely more those from underrepresented groups.

7. Given that first and last authorship are usually more prestigious, do you think future surveys should also ask question about the authorship position?

8. Methods - please specify if the survey was sent in English only.

Reviewer #2:

Amano et al. perform an impressive cross-country survey of 908 environmental scientists to untangle the relative importance of linguistic, economic, and gender backgrounds on scientific productivity. I recommend this manuscript for publication with a few minor suggestions for improvement.

1. The authors left out several crucial variables on the survey that would have clarified the results immensely. Both the job title of the respondent and the prestige of their workplace institution could have explained several of the effects from the paper. For instance, it can be that women are systematically excluded from tenure-track positions, where productivity can be higher due to running a lab. Or, women are excluded from elite institutions, where productivity is higher due to greater administrative support and smaller teaching load. These possible explanations should be included in the discussion.

2. The authors group together all of the countries within income brackets together in the regression analysis.I would recommend controlling for each country specifically as a fixed effect, to isolate the effects within countries. This would strengthen the gender analysis, since none of the hypothesized mechanisms about gender depend on the country.

3. In addition to inferential statistics such as the p-values and coefficient estimates, the authors should report predictive statistics. For instance, in performing (say) 10-fold cross-validation, how well does the model do at predicting a researcher's productivity, compared to an average baseline where everyone is predicted to produce the same number of papers.

Reviewer #3:

The manuscript tried to address an important topic—structural disparities in scientific productivity related to language, income, and gender. The authors present a well-organized analysis using survey data and regression analysis. This study analyzes survey data from 908 environmental scientists across eight countries to examine how language, economic, and gender disparities relate to scientific productivity. The authors find that non-native English speakers, scientists from lower-income countries, and women report significantly fewer peer-reviewed publications compared to their native English-speaking, high-income country, and male counterparts. The authors argue that these disparities reflect structural inequities rather than differences in researcher ability or effort.

However, there are several key limitations before publication in a high-impact venue like PLOS Biology. The question I have about the manuscript are quite basic.

First, the survey is limited to 908 environmental scientists from eight countries, and eligibility required authors to have at least one first-authored English-language publication. This introduces selection bias by excluding those who may have already exited academia due to systemic barriers—potentially underestimating disparities. The paper acknowledges this but does not sufficiently explore the implications for generalizability or suggest methods to address survivor bias in future research.

Second, the study uses country-level proxies for both income and English proficiency rather than individual-level metrics. While practical, this potentially masks important within-country heterogeneity. For example, defining non-native English speakers as those from countries with "low" proficiency may oversimplify nuanced linguistic experiences. Furthermore, individual characteristics are widely known to be correlated with research capabilities. In addition, it is also worth considering including country fixed effect.

Thirdly, although the authors caution against making causal claims, some interpretations appear causal in tone. Furthermore, the mechanisms driving the observed productivity gaps are not deeply explored. Are disparities due to time costs, access to networks, publishing fees, mentorship, or other structural issues? Even speculative discussion or supporting evidence from qualitative research would enrich the paper's explanatory value.

Fourth, the regression results are not presented in a professional or standardized format. I recommend referring to classical economics papers as a guide for reporting. For example, with respect to the gender variable, the male category appears to be the omitted baseline, but this should be clearly stated in the table notes or main text. Additionally, while there are five discipline categories, four coefficients are shown. It seems the omitted reference category is missing from the presentation and should be explicitly identified. Most importantly, a more straightforward and interpretable way to demonstrate the effects of gender, language, and income would be to visualize their predicted margins, after controlling for all other covariates in the model. This would provide a clearer sense of the magnitude and direction of these effects, making the results more accessible to readers.

Fifth, the number of publications is used as the sole metric to describe scientists' productivity. However, this is a rather limited measure. More meaningful indicators of a scientist's success should be considered—such as citation counts, number of papers published in top field journals, and other quality-based metrics. Incorporating these measures would provide a more robust and nuanced evaluation of scientific productivity.

Finally, the focus on environmental sciences and the geographic distribution of respondents (e.g., heavy representation from Japan) may skew findings. The authors should discuss the limits of disciplinary generalizability and whether similar patterns would hold in fields with different publishing cultures or collaboration norms.

The paper concludes with important implications for research assessment but lacks actionable policy suggestions. I hope these comments are helpful in the development of your manuscript.

---

## [Editor Report · Decision Letter 2]

15 Aug 2025

Dear Tatsuya,

Thank you for the submission of your revised Meta-Research Article "Language, economic and gender disparities widen the scientific productivity gap" for publication in PLOS Biology. On behalf of my colleagues and the Academic Editor, Ulrich Dirnagl, I'm pleased to say that we can in principle accept your manuscript for publication, provided you address any remaining formatting and reporting issues. These will be detailed in an email you should receive within 2-3 business days from our colleagues in the journal operations team; no action is required from you until then. Please note that we will not be able to formally accept your manuscript and schedule it for publication until you have completed any requested changes.

Sincerely, 

Roli

Senior Editor

PLOS Biology

rroberts@plos.org